# The Value of Histological Algorithms to Predict the Malignancy Potential of Pheochromocytomas and Abdominal Paragangliomas—A Meta-Analysis and Systematic Review of the Literature

**DOI:** 10.3390/cancers11020225

**Published:** 2019-02-15

**Authors:** Adam Stenman, Jan Zedenius, Carl Christofer Juhlin

**Affiliations:** 1Department of Oncology-Pathology, Karolinska Institutet, 171 76 Stockholm, Sweden; adam.stenman@ki.se; 2Department of Molecular Medicine and Surgery, Karolinska Institutet, 171 76 Stockholm, Sweden; jan.zedenius@ki.se; 3Department of Breast, Endocrine Tumours and Sarcoma, Karolinska University Hospital, 171 76 Stockholm, Sweden; 4Department of Pathology and Cytology, Karolinska University Hospital, 171 76 Stockholm, Sweden

**Keywords:** PASS, GAPP, histology, meta-analysis, paraganglioma, pheochromocytoma

## Abstract

Pheochromocytomas (PCCs) and abdominal paragangliomas (PGLs), collectively abbreviated PPGLs, are neuroendocrine tumors of the adrenal medulla and paraganglia, respectively. These tumors exhibit malignant potential but seldom display evidence of metastatic spread, the latter being the only widely accepted evidence of malignancy. To counter this, pre-defined histological algorithms have been suggested to stratify the risk of malignancy: Pheochromocytoma of the Adrenal Gland Scaled Score (PASS) and the Grading system for Adrenal Pheochromocytoma and Paraganglioma (GAPP). The PASS algorithm was originally intended for PCCs whereas the GAPP model is proposed for stratification of both PCCs and PGLs. In parallel, advances in terms of coupling overtly malignant PPGLs to the underlying molecular genetics have been made, but there is yet no combined risk stratification model based on histology and the overall mutational profile of the tumor. In this review, we systematically meta-analyzed previously reported cohorts using the PASS and GAPP algorithms and acknowledge a “rule-out” way of approaching these stratification models rather than a classical “rule-in” strategy. Moreover, the current genetic panorama regarding possible molecular adjunct markers for PPGL malignancy is reviewed. A combined histological and genetic approach will be needed to fully elucidate the malignant potential of these tumors.

## 1. Introduction

To correctly identify pheochromocytoma (PCC) and paraganglioma (PGL) patients with a future risk of disseminated disease is one of the clinical dilemmas that physicians and patients face regarding this disease. Although several preoperative parameters have been suggested as indicative of PCCs and PGLs (jointly referred to as PPGLs) with potential of aggressive behavior, the prognostication is foremost relying on the postoperative pathology report. Two separate histological prediction algorithms have been proposed as aiding tools in the distinction between benign and potentially malignant PPGLs: the Pheochromocytoma of the Adrenal Gland Scored Scale (PASS) [1] and the Grading System for Adrenal Pheochromocytoma and Paraganglioma (GAPP) [2].

The PASS algorithm was introduced in 2002 by Dr. Lester Thompson, and was originally designed for PCCs only. The model incorporates a total of 12 different histological features that are weighed with one or two points each based on the occurrence of these parameters in a pre-defined metastatic cohort from the original publication. These features include the occurrence of large nests/diffuse growth, central or confluent tumor necrosis, high cellularity, cellular monotony, tumor cell spindling, mitotic figures > 3/10 HPF, atypical mitotic figures, periadrenal adipose tissue invasion, vascular invasion, capsular invasion, profound nuclear pleomorphism and nuclear hyperchromasia. Examples of these histological parameters are presented in Figure 1. When applying a cut-off score of ≥4, the PASS algorithm correctly identified all 33 metastatic PCCs as no case scored <4—thereby ensuring a sensitivity of 100% and a risk stratification model with the potential to correctly “rule in” metastatic PCCs [1]. The PASS algorithm has since been assessed in several studies [3,4,5,6,7,8,9,10,11,12,13,14,15,16,17,18,19,20,21,22,23,24,25,26,27,28,29], of which numerous has verified the value of this approach and some which could not wholly reproduce the original findings. However, the algorithm also carries inter- and intraobserver variation, and the clinical value of the method has been debated [8,10].

The GAPP algorithm builds on the PASS model by incorporating four histological parameters from the latter and by adding immunohistochemical (Ki-67 index) and clinical (biochemical profile) data [2]. The GAPP model is designed for both PCCs and PGLs, and stratifies the PPGLs into three separate classes: a “well-differentiated type”, a “moderately differentiated type” and a “poorly differentiated type”, based on the scoring outcome. These types corresponded to patient prognosis, with excellent survival among “well-differentiated” PPGLs and poorer survival for the two latter groups. Moreover, PPGLs with high GAPP scores metastasized sooner than those with low scores. Although a younger algorithm in terms of numbers of studies reproducing the findings of the GAPP original publication, the resemblances between the PASS and GAPP algorithms supports the theory that the two models may result in similar outcomes in terms of predicting metastatic behavior. 

To our knowledge, no comprehensive meta-analysis of the PASS and GAPP scores in PPGLs has yet been reported. The objective of this review was therefore to summarize all studies performed on PPGLs in which the PASS and/or GAPP algorithms have been employed, and to visualize the overall sensitivity and specificity for the method to detect risk of future metastases. In addition, we review the current genetic advances within the field, with specific focus on the potential use of molecular aberrancies as adjunct markers of metastatic properties in PPGLs. 

## 2. Subjects and Methods

For the study selection, a PubMed search including the search terms ”PASS pheochromocytoma”, “PASS paraganglioma”, “GAPP pheochromocytoma” and “GAPP paraganglioma” was performed. Only original articles, letters and case reports detailing human material were included in this meta-analysis, thereby excluding preliminary (unpublished) posters, general review articles as well as any study conducted in animals. A separate search was also performed for “PASS glomus tumor” and “GAPP glomus tumor”, with zero returning references, and hence this meta-analysis was focused on sympathetic (abdominal) PGLs only. All published studies were manually scrutinized by one of the authors (CCJ), and judged suitable for inclusion if the study contained: (1) either a pheochromocytoma or abdominal paraganglioma cohort with (2) either a PASS or GAPP stratification and (3) clinical information regarding how malignancy was defined as well as (4) clearly identified subgroup information (number of tumor samples with and/or without evidence of synchronous or metachronous evidence of malignancy respectively, with the associated PASS and/or GAPP scores clearly identifiable). The study was not excluded if fully retrievable information was available for subsets of the cases, for example if the authors only presented data from metastatic tumors, or if only tumors with pathological scores were noted etc. No contact with study authors was performed in this process. The principal summary measures were to calculate an overall sensitivity, specificity, positive predictive value and negative predictive value for each tumor type and histological stratification system, combining the results of each individual study into master tables. To assess the bias risk of outlying individual studies, manuscripts were read in full to elucidate if the sample cohorts and histological algorithms used seemed to be adequately characterized and employed respectively. 

To widen the analysis beyond histology, we also reviewed the available scientific literature for genetic (DNA level) and expressional (RNA/protein level) markers that have been found valuable in distinguishing metastatic from non-metastatic PPGL, and summarize a selection of promising candidates with established or potential future value for screening purposes of clinical material.

## 3. Results

### 3.1. The PASS Algorithm: Study Selection

In total, 50 published studies were identified by the PubMed search engine when employing the term “PASS pheochromocytoma”. After an initial assessment of eligibility, including the presence of a PASS stratified pheochromocytoma cohort as well as an exclusion of irrelevant studies and reviews, 28 original studies and case reports were initially included in the review [1,3,4,5,6,7,8,9,10,11,12,13,14,15,16,17,18,19,20,21,22,23,24,25,26,27,28,29]. After careful revision of each study, eight additional studies were disqualified from inclusion based on the failure to meet our fourth inclusion criterion of “clearly identified subgroup information (number of tumor samples with and/or without evidence of synchronous or metachronous evidence of malignancy respectively, with the associated PASS score clearly identifiable)” [5,6,8,9,13,14,18,28]. Therefore, a total of 20 studies were included in the meta-analysis [1,3,4,7,10,11,12,15,16,17,19,20,21,22,23,24,25,26,27,29]. These studies and the associated outcomes are detailed in Table 1 and illustrated in Figure 2A. Using an identical approach for PGLs, eight studies in which the PASS algorithm was applied on this tumor type were originally identified [3,11,19,22,24,28,29,30], which was reduced to six after two publications [28,30] failed to meet the fourth inclusion criterion described above. In all, the survey period ranged from 2002 to 2018.

### 3.2. The PASS Algorithm: Meta-Analysis 

Altogether, in the meta-analysis of the PASS algorithm, 848 PCCs were included, of which 809 cases with clearly identifiable subgroup information for all or at least parts of the material. Of these cases, 105 (13%) were defined as malignant (Table 1). The definition of “malignant PCC” varied between studies, with the most common criterion consisting of “distant metastases only” (13 studies) followed by “metastatic disease or local recurrences” (four studies) and single studies with definitions of “metastatic disease or direct overgrowth onto adjacent organs” and “recurrence” respectively (Table 1). Of the 105 PCCs defined as malignant, 102 cases (97%) displayed a PASS score of ≥4, whereas only three malignant cases (3%) had PASS scores <4 (Table 1, Figure 2A). Among the benign PCCs, 224 cases exhibited a PASS score of ≥4 and 480 cases displayed PASS scores <4 (Figure 2A). The overall sensitivity for the PASS algorithm to correctly identify a malignant PCC was 97%, whereas the specificity was 68%. Given the rarity of malignant PCCs compared to the benign counterpart, the positive (PPV) and negative predictive values (NPV) for the PASS algorithm were calculated to obtain numbers in relation to prevalence. The PPV was 31% and the NPV 99%, meaning that a “positive finding” (PASS score ≥4) will not clearly indicate whether or not the PCC should be considered to carry a risk of malignant potential or not (Table 1). However, a “negative finding” (PASS score of <4) is highly indicative of a benign clinical course in this summarized material of PCC patients.

In addition to the PCCs, five studies reporting PASS scores from a total 56 PGLs were identified, of which 42 cases with clearly identifiable subgroup information for all or at least parts of the material (Table 2). All 13 malignant PGLs displayed PASS scores of ≥4, whereas no malignant cases had PASS scores <4 (Figure 2A). Among the benign PGLs, eight cases exhibited a PASS score of ≥4 and 21 cases displayed PASS scores <4. These results yielded a sensitivity of 100%, a specificity of 72%, a PPV of 62% and an NPV of 100%, indicating that PGL cases with PASS scores <4 are clinically benign (Table 2). 

### 3.3. The GAPP Algorithm: Study Selection and Meta-Analysis

For the GAPP algorithm, the results are presented in Table 3A for PCCs, in Table 3B for PGLs and schematically illustrated in Figure 2B. The survey period for the GAPP studies ranged from 2014 to 2018. Four studies reporting the GAPP scores for PCCs were originally identified [2,24,26,31], but only three met all our inclusion criteria and were included in this meta-analysis [2,24,26]. In total, out of 199 PCCs, GAPP scores were retrievable from 175 PCCs, of which four were malignant (2%) (Table 3A). Malignancy was defined by metastases (two studies) or metastases/local recurrences (one study). Of these four malignant cases, two exhibited GAPP scores ≥3, and two had scores <3 (Table 3A, Figure 2B). Of the benign PCCs, 35 cases displayed GAPP scores ≥3 and 136 cases scored <3. The corresponding sensitivity was 50%, the specificity was 80%, the PPV was 5% and the NPV was 99% (Table 3A). For PGLs, three studies with a total of 51 cases, of which 35 PGLs were included as all or subsets of data were available for these cases (Table 3B) [2,24,32]. Four cases were defined as malignant, and all of these displayed GAPP scores ≥3, whereas 10 and 21 benign PGLs showed GAPP scores of ≥3 and <3 respectively (Table 3B, Figure 2B). The sensitivity was 100% and the specificity was 68%, yielding a PPV of 29% and an NPV of 100%. 

### 3.4. Molecular Markers of Malignancy in PPGLs

From a genetic standpoint, PPGL carry the highest rate of heritability of all human tumors, with a growing list of well-characterized susceptibility genes that relate to a wide spectrum of pathways [33,34], which is also reflected by the numbers of reviews covering this topic. In short, a two-cluster system was originally suggested in 2011 based on an unsupervised mRNA expression analysis in PPGL [35]. Recently, a three-cluster molecular taxonomy of PPGL has been proposed based on the underlying mutations and altered pathways, including the pseudohypoxic PPGL (cluster 1), the Wnt signaling PPGL (cluster 2) and the kinase signaling PPGL (cluster 3—formerly cluster 2) [36]. In a recent, comprehensive analysis of The Cancer Genome Atlas (TCGA) database, genomic markers associated with metastatic disease (distant metastases, local recurrence or positive regional lymph nodes) included SDHB germline mutations, MAML3 fusion gene variants, somatic mutations in SETD2 or ATRX, a high number of somatic mutations in total, a hypermethylation subtype and the two mRNA subtypes: the Wnt-altered and the pseudohypoxia [37]. Moreover, TERT promoter mutations, structural rearrangements and telomerase activation has been described for PPGLs, with an overrepresentation in metastatic tumors—suggesting that cellular immortalization could be a central component for the metastatic process and a promising molecular marker for cases at risk of spread disease [38,39,40]. In addition, the mammalian Target Of Rapamycin (mTOR) pathway has been found dysregulated in metastatic PPGLs and display activation in tumors associated to SDHx gene mutations [41,42]. 

In a recent study, low expression of Chromogranin B (CHGB) (mRNA and protein) was associated with both PASS scores, occurrence of metastatic disease and shorter disease-related survival, suggesting CHGB as a possible marker for pinpointing PPGL with high PASS scores and aggressive tumor behavior [29].

## 4. Discussion

PPGLs are potentially curable by surgery, and the chances for remedy improve if the tumor is localized to the primary site and excised with negative margins. Disseminated disease however, is still difficult to treat based on limited treatment options with suboptimal effect. A patient with a resected PPGL without clinical evidence of metastatic disease is subject to clinical follow-up, and various parameters (from genetic to histological findings) could affect the duration as well as the interval of the follow-up screening. To be able to pinpoint cases at risk of future metastases directly postoperatively would therefore have a significant clinical impact, and great efforts have therefore been made trying to identify histological features that could predict the outcome of this patient category. 

In this first meta-analysis of histological prediction of PPGLs, we found that the PASS algorithm exhibited a fairly low PPV but an exceedingly high NPV for both PCCs and PGLs, indicating that this model is excellent in ruling out—rather than ruling in—malignant potential for both tumor types. Therefore, the true value of the PASS algorithm could be to pinpoint cases with an exceptionally low risk of future metastases, rather than to primarily identify cases at risk of disseminated disease. Moreover, it seems as though this model is excellent also for ruling out malignant potential in abdominal PGLs, in addition to PCCs for which the algorithm was primarily constructed. Although based on a much smaller material, similar findings were seen when analyzing the GAPP algorithm—with high NPVs for both PCCs and PGLs, suggesting a “rule-out” function for this model as well. 

The similar results obtained by the PASS and GAPP algorithms are intriguing, since the latter model expands on the former by also including immunohistochemistry and biochemical data. On the other hand, the PASS algorithm displays a greater number of histological criteria included. As both scoring systems displayed equally excellent NPVs, it seems likely that the fewer histological criteria covered by the GAPP algorithm in theory is compensated by the addition of counting a Ki-67 index and evaluating the catecholamine profile of the tumor.

A limitation to the current study is the fact that our data stems from several unique reports obtained from separate pathology departments, which could affect the overall results—not least given the previously established observer variation of the PASS algorithm. This is particularly evident in terms on how the various study authors defined malignancy, which could lead to conflicting results when interpreting the outcome of our meta-analysis. However, as the majority of studies included metastatic disease as the sole criterion for malignancy, we believe that our results closely reflect the potential for the PASS and GAPP algorithms to detect metastatic potential. It should be noted however, that subsets of studies also defined malignancy by local recurrences—a biological phenomenon which could pinpoint locally aggressive tumors, but not necessarily PPGLs with metastatic potential. Furthermore, since PPGLs are rare tumors, this reflects the rather scarce number of comprehensive histological reports available across the literature. There was a rather substantial heterogeneity in terms of how the cohorts were presented, and several studies did not properly define whether the metastases were detected syn- or metachronously. Moreover, the follow-up period varied greatly between different studies, but also within the same study cohorts, with several reports presenting follow-up time for individual cases ranging from <1 year up to 20–30 years. To exclude individual studies (or individual cases within studies) on the basis of heterogeneous follow-up time alone is certainly a matter of debate, but nevertheless would put a heavy strain on the number of included studies in total.

Moreover, the general limitations of each meta-analysis also apply here, such as publication bias (“positive” results are more likely to get published), search and selection bias (identifying the correct studies) as well as the heterogeneity of the results obtained (exemplified by the varied levels of standard in how the results were presented and the associated data could be extracted). Another practical limitation to our results from this meta-analysis is the rather large observer variability, which has been previously demonstrated for the PASS system [8]. Indeed, interobserver variability is frequently reported for various histological algorithms with fewer factors to consider than the PASS algorithm, thereby demonstrating the universal difficulty with visual interpretation of objective parameters [43,44,45].

The idea that the PASS and GAPP algorithms display excellent sensitivity towards malignant disease is possibly due to the fact that both systems embrace many different histological parameters traditionally constituting “rule in” criteria for malignant disease in general. However, as a consequence of this “histological shotgun” approach, the specificity is reduced significantly. To obtain algorithms with superior specificity (and thereby also an improved positive predictive value), it is most likely that we need to turn to additional immunohistochemical and/or molecular genetic markers for this purpose. Indeed, previous reports have found the inclusion of sustentacular cell counts by S100 immunohistochemistry, estimation of the Ki-67 proliferation index and overall tumor size to be helpful additions to the current PASS algorithm [12,16].

Moreover, a large number of markers with various level of evidence have been proposed as genetic and expressional indicators of metastatic disease. Significant markers include constitutional mutations in *SDHB*, somatic mutations in *ATRX* and *SETD2* or a high somatic mutational burden in total, *TERT* gene abberancies, gene fusions involving *MAML3,* a hypermethylation subtype and the two mRNA clustering subtypes: the pseudohypoxia (cluster 1) and the Wnt-altered (cluster 2). In addition, low tumoral and plasma levels of CHGB has recently been suggested as a marker for potential aggressive behavior. It is possible that a study combining several of the abovementioned genetic and/or expressional aberrations will display PPVs superior to that of both the PASS and GAPP algorithms, so that a future assessment of a PPGL tumor would require both histological and molecular investigations to reach the ultimate endpoint of a classification system with near-perfect PPV and NPV. However, many of these markers—although promising—need to be clinically validated in large, prospective studies before being considered for future inclusions in a combined histology-molecular genetics type of algorithm. 

As most of the molecular markers detailed above are based on investigations from surgically resected PPGL specimen, future studies regarding the potential usage of these markers in a liquid biopsy setting (for example circulating tumor cells and/or cell-free tumor DNA) could be of value. This is especially true for patients with multiple tumors detected at diagnosis—as histological analyses might not be possible to perform on each individual lesion. Indeed, the potential of detecting molecular aberrancies signifying metastatic potential in a non-invasive manner in the pre-operative setting would be the ultimate end-point for any prognostic marker.

We conclude that the PASS and GAPP algorithms could be used for the assessment of metastatic potential in PPGLs, but the interpretation of the results should probably be focused around a “rule-out” way of thinking rather than the traditional “rule-in” approach. Low scores would strongly imply a benign clinical course, whereas high scores leave the clinician with little valuable information regarding future risks. Moreover, next-generation sequencing data identifying pathogenic mutations within cluster 1 or 2 could in theory improve the “rule-in” aspect of the prognostication, but this approach needs to be validated in larger tumor cohorts. In all, a comprehensive molecular approach will probably be needed to cover up the inability of the current histological algorithms to accurately pinpoint cases at risk for future metastases.

## 5. Conclusions

The current histological systems available for grading malignant potential in pheochromocytoma and abdominal paraganglioma should primarily be used as rule-out algorithms, pinpointing cases with exceedingly low risks of future metastases.

## Figures and Tables

**Figure 1 cancers-11-00225-f001:**
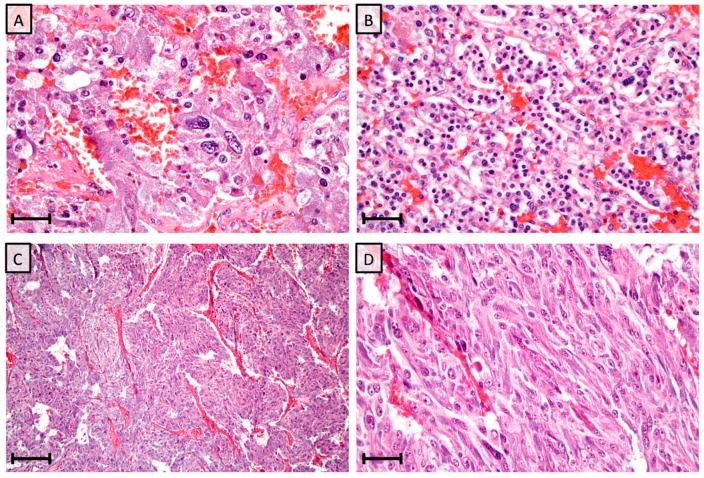
Photomicrographs of metastatic (**A**,**B**) and non-metastatic (**C**,**D**) pheochromocytoma cases with elevated PASS scores previously diagnosed at our institution. Scale bars are 25 micrometers for **A**,**B** and **D**, and 100 micrometers for **C**. (**A**) Nuclear pleomorphism in a pheochromocytoma with a total PASS score of 8. This tumor was resected from a 61-year old female who developed metastatic disease 9 years after initial diagnosis. (**B**) Same case displaying hypercellularity and nuclear hyperchromasia, two additional parameters included in the PASS algorithm. (**C**) Large and irregular nests in a pheochromocytoma with a PASS score of 7, diagnosed in a 41-year old male. The patient is alive without metastatic disease after 20 years of follow-up. (**D**) Same case displaying focal tumor cell spindling with elongated nuclei, a phenomenon yielding two PASS points.

**Figure 2 cancers-11-00225-f002:**
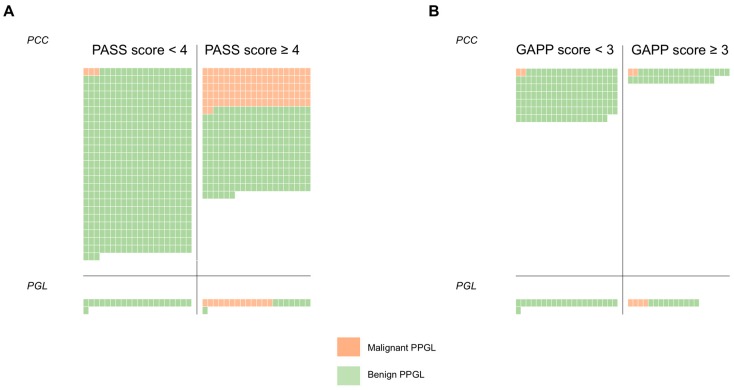
Schematic overview of the (**A**) PASS and (**B**) GAPP meta-analyses outcome in pheochromocytoma (PCC) and abdominal paraganglioma (PGL). Each tumor sample is represented by a square, in which green color indicates a benign tumor as according to the definition by each study. Orange squares denote cases defined as malignant. The left column of each classification system signifies number of cases with low algorithm scores, whereas the right column contains cases with scores ≥4 (PASS) and ≥3 (GAPP). As demonstrated here, both algorithms exhibit excellent sensitivity but reduced specificity towards malignant cases. These analyses indicate that low PASS and GAPP scores almost always are associated with a benign clinical course.

**Table 1 cancers-11-00225-t001:** PCC cohorts stratified by the PASS algorithm.

Study No.	First Author	Year Published	Number of PCCs *	Number of Malignant PCCs *	Definition of Malignant PCCs	Mal PCCs PASS ≥ 4	Mal PCCs PASS < 4	Benign PCCs PASS ≥ 4	Benign PCCs PASS < 4	SENS	SPEC	PPV	NPV
1	Thompson	2002	100	33	MET	33	0	17	50	100%	75%	66%	100%
2	August	2004	37	14	MET	14	0	23	0	100%	0%	38%	0%
3	Kajor	2005	40	1	MET	1	0	7	32	100%	82%	13%	100%
4	Strong	2008	47	5	MET	5	0	10	32	100%	76%	33%	100%
5	Agarwal	2010	90	6	MET/DO	5	1	27	57	83%	68%	16%	98%
6	Szalat	2010	26	7	MET	6	1	0	19	86%	100%	100%	95%
7	de Wailly	2012	21	7	MET	7	0	7	7	100%	50%	50%	100%
8	Mlika	2013	11	2	MET	2	0	6	3	100%	33%	25%	100%
9	Bialas	2013	62	5	REC/MET	5	0	29	28	100%	49%	15%	100%
10	Ocal	2014	11	3	REC	3	0	4	4	100%	50%	43%	100%
11	Kulkarni	2016	6	1	MET	1	0	2	3	100%	60%	33%	100%
12	Lupşan	2016	17	13	MET	13	0	2	2	100%	50%	87%	100%
13	Suenaga	2016	1	0	REC	0	0	1	0	*npd*	*npd*	*npd*	*npd*
14	Kim	2016	90	*ns*	REC/MET	*npd*	0	*npd*	52	*npd*	*npd*	*npd*	*npd*
15	Maignan	2017	65	0	MET	0	0	9	56	*npd*	86%	*npd*	*npd*
16	Koh	2017	32	4	MET	3	1	19	9	75%	32%	14%	90%
17	Aggeli	2017	69	0	MET	*ns*	*ns*	31	37	*npd*	54%	*npd*	*npd*
18	Stenman	2018	41	0	REC/MET	0	0	10	31	*npd*	76%	*npd*	*npd*
19	Muchuweti	2018	1	0	MET	0	0	1	0	*npd*	*npd*	*npd*	*npd*
20	Stenman	2018	81	4	REC/MET	4	0	19	58	100%	75%	17%	100%
**Summarized**		-	**848**	**105**	-	**102**	**3**	**224**	**480**	**97%**	**68%**	**31%**	**99%**

MET—metastatic disease, REC—recurrence, DO—direct overgrowth, ns—not specified, npd—not possible to determin, SENS—sensitivity, SPEC—specificity, PPV—positive predictive value, NPV—negative predictive value; *—Numbers correspond to cases histologically investigated, which is not necessarily identical to cases included in the study as a whole. Numbers in bold script at the bottom represent summarized values for all parameters, with corresponding SENS, SPEC, PPV and NPV values calculated for these sums.

**Table 2 cancers-11-00225-t002:** PGL cohorts stratified by the PASS algorithm.

Study No.	First Author	Year Published	Number of PGLs *	Number of Malignant PGLs *	Definition of Malignant PGLs	Mal PGLs PASS ≥ 4	Mal PGLs PASS < 4	Benign PGLs PASS ≥ 4	Benign PGLs PASS < 4	SENS	SPEC	PPV	NPV
1	August	2004	6	6	MET	6	0	0	0	100%	-	100%	-
2	Szalat	2010	1	1	MET	1	0	0	0	100%	-	100%	-
3	Kulkarni	2016	4	2	MET	2	0	0	2	100%	100%	100%	100%
4	Kim	2016	29	16	REC/MET	*npd*	0	*npd*	15	*npd*	*npd*	*npd*	*npd*
5	Koh	2017	5	0	MET	0	0	3	2	*npd*	*npd*	*npd*	*npd*
6	Stenman	2018	11	4	REC/MET	4	0	5	2	100%	29%	44%	100%
**Summarized**		-	**56**	**29**	-	**13**	**0**	**8**	**21**	**100%**	**72%**	**62%**	**100%**

PGL—paraganglioma, MET—metastatic disease, REC—recurrence, ns—not specified, npd—not possible to determine, SENS -sensitivity, SPEC—specificity; ns -not specified, npd—not possible to determine, PPV—positive predictive value, NPV—negative predicitive value; *—Numbers correspond to cases histologically investigated, which is not necessarily identical to cases included in the study as a whole. Numbers in bold script at the bottom represent summarized values for all parameters, with corresponding SENS, SPEC, PPV and NPV values calculated for these sums.

**Table 3 cancers-11-00225-t003:** PPGL cohorts stratified by the GAPP algorithm.

Study No.	First Author (Year Published)	Number of PCCs	Number of Malignant PCCs *	Definition of Malignant PCCs *	Mal PCCs GAPP ≥ 3	Mal PCCs GAPP < 3	Benign PCCs GAPP ≥ 3	Benign PCCs GAPP < 3	SENS	SPEC	PPV	NPV
**A. PCC cohorts stratified by the GAPP algorithm.**												
1	Kimura (2014)	126	24	MET	*npd*	*npd*	0	102	*npd*	*npd*	*npd*	*npd*
2	Koh (2017)	32	4	MET	2	2	19	9	50%	32%	10%	82%
3	Stenman (2018)	41	0	REC/MET	0	0	16	25	*npd*	61%	*npd*	*npd*
**Summarized**	-	**199**	**28**	-	**2**	**2**	**35**	**136**	**50%**	**80%**	**5%**	**99%**
**B. PGL cohorts stratified by the GAPP algorithm.**												
1	Kimura (2014)	36	16	MET	*npd*	*npd*	0	20	*npd*	*npd*	*npd*	*npd*
2	Gupta (2016)	10	4	MET	4	0	6	0	100%	0%	40%	*npd*
3	Koh (2017)	5	0	MET	0	0	4	1	*npd*	*20%*	*npd*	*npd*
**Summarized**	-	**51**	**20**	-	**4**	**0**	**10**	**21**	**100%**	**68%**	**29%**	**100%**

PCC—pheochromocytoma, PGL—paraganglioma, MET—metastatic disease, REC—recurrence; MET—metastatic disease, REC—recurrence, ns - not specified, npd—not possible to determine; SENS—sensitivity, SPEC—specificity, PPV—positive predictive value; PPV—positive predictive value, NPV—negative predicitive value; *—Numbers correspond to cases histologically investigated, which is not necessarily identical to cases included in the study as a whole. Numbers in bold script at the bottom represent summarized values for all parameters, with corresponding SENS, SPEC, PPV and NPV values calculated for these sums.

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
