# Peer review of "The Value of Histological Algorithms to Predict the Malignancy Potential of Pheochromocytomas and Abdominal Paragangliomas—A Meta-Analysis and Systematic Review of the Literature"

_cancers, 2019, doi:10.3390/cancers11020225_

Reviewer 1 Report

Stenman et al revisited published papers on the application of histological algorithms PASS (Pheochromocytoma of the Adrenal Gland Scaled Score) and GAPP (The Grading system for Adrenal Pheochromocytoma and Paraganglioma) with the aim to assess the value of these score systems to predict the malignant potential of pheochromocytomas and paragangliomas. They pointed out a “rule-out” way of stratification models rather than a classical “rule-in” strategy. In addition, they suggest that a combined histological and genetic approach will be needed to fully elucidate the malignant potential of these tumors.

To the best of my knowledge this is the first time that a meta-analysis is intended to evaluate the aforementioned histological algorithms. Therefore, I found the study interesting, sound and timely. Below some minor comments:

1. The definition of malignant PPGL has changed according to the most recent World Health Organization (WHO) classification (Lloyd RV et al. WHO classification of tumours: pathology and genetics of tumours of endocrine organs. Lyon: IARC; 2017). Therefore this definition should be used throughout the manuscript.

2. Related with the previous comment, it should be stressed the necessity to better distinguish aggressive from readily metastatic PPGL which may lead to confusion in the discussed studies (aggressive are not necessarily metastatic).

3. Another aspect to take into account as criteria to include those studies is whether they satisfy a reasonable follow-up period of metastatic patients and whether they state that metastases were synchronous or metachronous.

4. The authors proposed to review the emergent molecular markers of metastatic PPGL. However some of those markers did not appear in the text (mTOR and TERT for instance) whereas others have not been clinically validated in large, multi-centric and prospective studies and for the moment correspond to correlative observations rather to formal analyses related with survival parameters and risk assessment (MAML3 for instance). I suggest to stress this important point. In fact, I agree that a combination of molecular markers and the algorithms would be very useful but awaits formal confirmatory analyses.

5. Another aspect of metastatic PPGL is tumor heterogeneity. Some patients have multiple tumors by the moment of diagnosis and it is impractical to perform histological analyses for all of them. Authors should comment on this inconvenience for histological diagnosis using those algorithms and the possibility to go move forward into the analysis of liquid biopsies for the detection of relevant molecular markers.

6. It is also convenient to make a comment on how to improve the histological assessment of such algorithms. For instance, the study cited in ref 16 (Białas M et al) showed that the number of sustentacular cells (S-100+ cells) is lower in metastatic compared to benign PPGL. Why such marker has not been implemented in the PASS score for better assessment of tissue architecture?

Author Response

Ms. Treena Guo

Assistant Editor, Cancers        

Stockholm, February 11th 2019

Dear Ms. Guo,

Please find attached our revised review article intended for the upcoming special issue on "Pheochromocytoma (PHEO) and Paraganglioma (PGL)" entitled “The value of histological algorithms to predict malignant potential of pheochromocytomas and abdominal paragangliomas – a meta-analysis and systematic review of the literature” by Adam Stenman, Jan Zedenius and C. Christofer Juhlin.

We want to extend our sincere gratitude to you and the two reviewers for considering our manuscript, and for suggesting changes that greatly have improved the overall study.

We have made corrections to the main text and Figure 1 to meet the requirements and suggestions proposed by each referee. All changes to the main text have been marked up in yellow for clarity. The specific responses to the comments raised by the Editor as well as Reviewer 1 follow below. Our replies to Reviewer 2 are detailed in a separate document.

Editorial comment:

”Thank you for submitting your manuscript. It has been reviewed by experts in the field and we request that you make minor revisions before it is processed further.”

Reply: We thank the Editor for this positive remark. All reviewer queries have been addressed.

Reviewer 1:

”Stenman et al revisited published papers on the application of histological algorithms PASS (Pheochromocytoma of the Adrenal Gland Scaled Score) and GAPP (The Grading system for Adrenal Pheochromocytoma and Paraganglioma) with the aim to assess the value of these score systems to predict the malignant potential of pheochromocytomas and paragangliomas. They pointed out a “rule-out” way of stratification models rather than a classical “rule-in” strategy. In addition, they suggest that a combined histological and genetic approach will be needed to fully elucidate the malignant potential of these tumors.

 To the best of my knowledge this is the first time that a meta-analysis is intended to evaluate the aforementioned histological algorithms. Therefore, I found the study interesting, sound and timely. Below some minor comments:”

1. ”The definition of malignant PPGL has changed according to the most recent World Health Organization (WHO) classification (Lloyd RV et al. WHO classification of tumours: pathology and genetics of tumours of endocrine organs. Lyon: IARC; 2017). Therefore this definition should be used throughout the manuscript.”

Reply: We agree that the nomenclature should be in line with the 2017 WHO criteria, namely “metastatic” and “non-metastatic” PPGL. We have therefore revised the manuscript and made several changes throughout whenever possible. Please note that many studies included in this meta-analysis were published well ahead of the release of the current WHO classification, and therefore subsets of authors defined malignancy also by local recurrences and direct overgrowth. We have therefore tried to change the nomenclature in the main text body and our discussion, without interfering with some of the authors original definitions that are detailed in the tables and the Results section.

2. “Related with the previous comment, it should be stressed the necessity to better distinguish aggressive from readily metastatic PPGL which may lead to confusion in the discussed studies (aggressive are not necessarily metastatic).”

Reply: We concur that the terminology could be confusing. A limitation to our study is the fact that our data is obtained from separate pathology departments. This is particularly evident in terms on how the various study authors defined malignancy, which could lead to conflicting results when interpreting the outcome of our meta-analysis. However, as the majority of studies included metastatic disease as the sole criterion for malignancy, we believe that our results closely reflect the potential for the PASS and GAPP algorithms to detect metastatic potential. It should be noted however, that subsets of studies also defined malignancy by local recurrences – a biological phenomenon which could pinpoint locally aggressive tumors, but not necessarily PPGLs with metastatic potential.

We have added a passage of text to the Discussion section in which this issue is brought up.

3. “Another aspect to take into account as criteria to include those studies is whether they satisfy a reasonable follow-up period of metastatic patients and whether they state that metastases were synchronous or metachronous.”

Reply: This is a great point that also highlights another limitation to our meta-analysis, and we did in fact discuss these inclusion criteria during the data collection phase. Since PPGLs are rare tumors, this reflects the rather scarce number of comprehensive histological reports available across the literature. There was a rather substantial heterogeneity in terms of how the cohorts were presented, and several studies did not properly define whether the metastases were detected syn- or metachronously. Moreover, the follow-up period varied greatly between different studies, but also within the same study cohorts, with several reports presenting follow-up time for individual cases ranging from<1 year up to 20-30 years. To exclude individual studies (or individual cases within studies) on the basis of heterogeneous follow-up time alone is certainly a matter of debate, but nevertheless would put a heavy strain on the number of included studies in total.

 We have added a passage of text to the Discussion section in which this issue is brought up.

4. “The authors proposed to review the emergent molecular markers of metastatic PPGL. However some of those markers did not appear in the text (mTOR and TERT for instance) whereas others have not been clinically validated in large, multi-centric and prospective studies and for the moment correspond to correlative observations rather to formal analyses related with survival parameters and risk assessment (MAML3 for instance). I suggest to stress this important point. In fact, I agree that a combination of molecular markers and the algorithms would be very useful but awaits formal confirmatory analyses.”

Reply: We thank the referee for this excellent point. We have revised this part of the manuscript by adding mTOR and TERT to the list of promising molecular markers of metastatic disease, and also defined the need for our proposed markers to be clinically validated in prospective studies before being considered for future inclusions in a combined histology-molecular genetics type of algorithm. Moreover, additional references have been added, and the reference list has been updated accordingly.

5. “Another aspect of metastatic PPGL is tumor heterogeneity. Some patients have multiple tumors by the moment of diagnosis and it is impractical to perform histological analyses for all of them. Authors should comment on this inconvenience for histological diagnosis using those algorithms and the possibility to go move forward into the analysis of liquid biopsies for the detection of relevant molecular markers.”

Reply: Another great point. We have added a passage in the Discussion section in which this topic is addressed.

6. “It is also convenient to make a comment on how to improve the histological assessment of such algorithms. For instance, the study cited in ref 16 (Białas M et al) showed that the number of sustentacular cells (S-100+ cells) is lower in metastatic compared to benign PPGL. Why such marker has not been implemented in the PASS score for better assessment of tissue architecture?”

Reply: We agree that there might be several modifications to the original PASS algorithm that would improve the sensitivity and specificity even more, including the role of an S100 staining. The Discussion section of the revised manuscript now contains a passage of text detailing this matter.

We thank the referees for suggesting substantial improvements to our manuscript. We hope that the Editors and reviewers find the above-suggested changes to be in line with their intentions and will find our manuscript of sufficient quality to warrant publication.

Best regards,

Carl Christofer Juhlin, MD, Associate Professor

Dept. of Oncology-Pathology, Karolinska Institutet, Stockholm, Sweden

Reviewer 2 Report

The review of Stenman and coworkers summarized the current literature on histological algorithms to predict the malignant potential of PPGLs. Furthermore, they performed a systematic meta-analysis of previously reported cohorts and recommend the preference of a “rule-out” strategy rather than the classical “rule-in” system. The manuscript is well written, but some small comments should be addressed.

Abstract:

Line 27: remove “screened”

Line 31: Please remove “enigmatic” à unscientific term in this context

Figure 1:

Please add a scale bar

Subjects and Methods:

Please add the survey period (oldest and youngest publication) in the text

Line 105-108: In the results section you only described the three main cluster and the associated genes in PPGL not as indicated in the “Subjects and Methods” section mentioned expressional and genetic markers. This would be quite interesting. Please comment and make it more clearly in the text. Are there other markers described, besides CHGB?

Results:

Line 113: remove the second “

In your meta-analysis the GAPP algorithm seems not to have any advantage in comparison to PASS. This is unsuspected, at least for me, because you provide more information to the scoring. What is your explanation for that? Please discuss it in the manuscript.

Author Response

Ms. Treena Guo

Assistant Editor, Cancers        

Stockholm, February 11th 2019

Dear Ms. Guo,

Please find attached our revised review article intended for the upcoming special issue on "Pheochromocytoma (PHEO) and Paraganglioma (PGL)" entitled “The value of histological algorithms to predict malignant potential of pheochromocytomas and abdominal paragangliomas – a meta-analysis and systematic review of the literature” by Adam Stenman, Jan Zedenius and C. Christofer Juhlin.

We want to extend our sincere gratitude to you and the two reviewers for considering our manuscript, and for suggesting changes that greatly have improved the overall study.

We have made corrections to the main text and Figure 1 to meet the requirements and suggestions proposed by each referee. All changes to the main text have been marked up in yellow for clarity. The specific responses to the comments raised by the Editor as well as Reviewer 2 follow below. Our replies to Reviewer 1 are detailed in a separate document.

Editorial comment:

”Thank you for submitting your manuscript. It has been reviewed by experts in the field and we request that you make minor revisions before it is processed further.”

Reply: We thank the Editor for this positive remark. All reviewer queries have been addressed.

Reviewer 2:

”The review of Stenman and coworkers summarized the current literature on histological algorithms to predict the malignant potential of PPGLs. Furthermore, they performed a systematic meta-analysis of previously reported cohorts and recommend the preference of a “rule-out” strategy rather than the classical “rule-in” system. The manuscript is well written, but some small comments should be addressed.”

Abstract:

1. “Line 27: remove “screened””

Reply: The word has been deleted.

2. “Line 31: Please remove “enigmatic” Ă  unscientific term in this context”

Reply: We agree. The word has been deleted.

Figure 1:

3. “Please add a scale bar”

Reply: Scale bars have been added to the photomicrographs in Figure 1 as suggested.

Subjects and Methods:

4. “Please add the survey period (oldest and youngest publication) in the text”

Reply: The survey periods for the PASS and GAPP algorithms were 2002-2018 and 2014-2018 respectively. This information has been added to the Results section of each scoring algorithm.

5. “Line 105-108: In the results section you only described the three main cluster and the associated genes in PPGL not as indicated in the “Subjects and Methods” section mentioned expressional and genetic markers. This would be quite interesting. Please comment and make it more clearly in the text. Are there other markers described, besides CHGB?”

Reply: We concur that this part of the manuscript could be expanded. We have revised this part of the manuscript by adding mTOR and TERT to the list of promising molecular markers of metastatic disease, and also defined the need for our proposed markers to be clinically validated in prospective studies before being considered for future inclusions in a combined histology-molecular genetics type of algorithm. Moreover, additional references have been added, and the reference list has been updated accordingly.

Results:

6. “Line 113: remove the second “ “

Reply: We thank the referee for spotting this typo. The quotation mark has been deleted.

7. “In your meta-analysis the GAPP algorithm seems not to have any advantage in comparison to PASS. This is unsuspected, at least for me, because you provide more information to the scoring. What is your explanation for that? Please discuss it in the manuscript.”

Reply: This is a great point, and we agree that the inclusion of immunohistochemistry as well as biochemical data would increase the validity of GAPP compared to PASS – although the latter algorithm display a greater number of histological criteria included. As both scoring systems displayed equally excellent NPVs, it seems likely that the fewer histological criteria covered by the GAPP algorithm is compensated by the addition of counting a Ki-67 index and evaluating the catecholamine profile of the tumor. We have added a paragraph to the Discussion section of the manuscript in which this topic is covered.

We thank the referees for suggesting substantial improvements to our manuscript. We hope that the Editors and reviewers find the above-suggested changes to be in line with their intentions and will find our manuscript of sufficient quality to warrant publication.

Best regards,

Carl Christofer Juhlin, MD, Associate Professor

Dept. of Oncology-Pathology, Karolinska Institutet, Stockholm, Sweden
